# Development and Evaluation of the Clinical Trial HEalth Knowledge and Beliefs Scale (CHEKS)

**DOI:** 10.3390/ijerph19148660

**Published:** 2022-07-16

**Authors:** Alicia Chung, Tiffany Donley, Ron D. Hays, Rebecca Robbins, Azizi Seixas, Girardin Jean-Louis

**Affiliations:** 1Center for Early Childhood Health and Development, Department of Population Health, NYU School of Medicine, 227 E. 30th Str., New York, NY 10036, USA; 2Institute for Excellence in Health Equity, Department of Population Health, NYU School of Medicine, 180 Madison Ave, New York, NY 10016, USA; tiffany.donley@nyulangone.org4; 3UCLA Department of Medicine, 1100 Glendon Avenue, Los Angeles, CA 90024, USA; drhays@ucla.edu; 4Division of Sleep Medicine, Harvard Medical School, Boston, MA 02115, USA; rrobbins4@bwh.harvard.edu; 5Division of Sleep and Circadian Disorders, Department of Medicine, Brigham & Women’s Hospital, Boston, MA 02115, USA; 6Psychiatry and Behavioral Sciences, University of Miami School of Medicine, Miami, FL 33136, USA; azizi.seixas@miami.edu (A.S.); girardin.jean-louis@miami.edu (G.J.-L.)

**Keywords:** clinical trial measure, assessment, clinical trial knowledge, evaluation, scale development, evaluation

## Abstract

Patient health literacy is vital to clinical trial engagement. Knowledge and beliefs about clinical trials may contribute to patient literacy of clinical trials, influencing engagement, enrollment and retention. We developed and assessed a survey that measures clinical trial health knowledge and beliefs, known as the **C**linical trial **HE**alth **K**nowledge and belief **S**cale (CHEKS). The 31 survey items in CHEKS represent knowledge and beliefs about clinical trial research (*n* = 409) in 2017. We examined item-scale correlations for the 31 items, eliminated items with item-scale correlations less than 0.30, and then estimated internal consistency reliability for the remaining 25 items. We used the comparative fit index (CFI) and the root mean squared error of approximation (RMSEA) to evaluate model fit. The average age of the sample was 34 (SD = 15.7) and 48% female. We identified 6 of the 31 items that had item-scale correlations (corrected for overlap) lower than 0.30. Coefficient alpha for the remaining 25 items was 0.93 A one-factor categorical confirmatory factor analytic model with 16 correlated errors was not statistically significant (chi-square = 10011.994, df = 300, *p* < 0.001) but fit the data well (CFI = 0.95 and RMSEA = 0.07). CHEKS can assess clinical trial knowledge and beliefs.

## 1. Introduction

Clinical trial participation is one of the largest challenges in healthcare research [1]. Limited clinical trial knowledge, including the understanding of research beyond oncology or pharmacological studies, contributes to negative beliefs and attitudes towards research [2]. Clinical trials are not limited to just pharmacological studies, but also include epidemiological, behavioral, health-services and community-engaged research, that aim to improve lifestyle behaviors for individual or community health outcomes. Yet, patients may not be aware of this breadth or variety in clinical trial research beyond pharmacology. Research shows that patient health literacy, which refers to the ability to understand and act upon health information in order to successfully navigate the healthcare environment [3], is one of the key determinants of clinical trial enrollment [4]. Health literacy is the degree in which a person understands health information, and demonstrates the skills to act and make informed decisions. Understanding an individual’s level of knowledge about clinical trials may help determine gaps that need to be addressed, to contribute to increased literacy levels. Psychometrically sound survey tools that assess clinical trial knowledge are absent from the literature. 

Unfortunately, patient health literacy is relatively low among adults in the United States. Specifically, only 12% of Americans demonstrate adequate health literacy [5], whereas approximately 50% of adults in Europe demonstrate adequate health literacy [6]. Additionally, poor patient health literacy may lead to poor disease management and overall health status [7]. Most clinical trial knowledge, informed consent, or literacy assessments focus on cancer clinical trials [8]. Existing health literacy tools in research, training or program planning, include the Short Assessment of Health Literacy—Spanish and English, Rapid Estimate of Adult Literacy in Medicine—Short Form, and Short Assessment of Health Literacy for Spanish Adults [8,9,10]. These literacy instruments provide clinicians and researchers with an assessment of a person’s ability to read, comprehend, understand and pronounce common health-related terms. Unfortunately, these instruments do not assess a patient’s knowledge or beliefs of the different types of clinical trial research, to guide researchers and clinicians on how to overcome barriers to patient engagement. We aimed to address this gap by developing an instrument for evaluation of clinical trial knowledge and beliefs, that was not oncology or pharmacological based. We sought to create a tool that assessed patient clinical trial knowledge of observational and community-based research instead, as well as knowledge about safeguards in place to protect patient safety and patient rights, to identify how much people knew about the structures in place to overcome typical barriers to research engagement. 

### History of Mistrust and Barriers to Participation

Unethical research conducted in human populations throughout history has contributed to grave mistrust, particularly among Black, Indigenous, and People of Color (BIPOC). Moreover, institutional racism, and lack of transparency, awareness, culturally and linguistically appropriate research measures, tools, and research studies are a few of the driving factors behind BIPOC patient mistrust and low rates of clinical trial participation [11]. Choi and colleagues found that fear, lack of trust, and patient safety concerns were barriers to clinical trial research patient enrollment in Korea [12]. This includes an understanding of patient clinical trial knowledge on such topics as patient rights, the safeguards in place to protect them, and the different types of research opportunities available. Unfortunately, only one clinical trial survey to date evaluates participant understanding about clinical trial research. The assessment awareness and perceptions of clinical trials survey in India found that education and awareness were key to addressing knowledge gaps [13]. The 10-item survey assesses ethical concerns and basic understanding of clinical trials but is limited in its evaluation of the participant understanding of the process of clinical trial research, safety safeguards, participants’ rights during research, and the potential benefits of community-based research with people of color. This survey tool was also weak in its methodological design, with a lack of inter-correlations and scale internal consistency reported. 

High patient knowledge has been positively associated with willingness to join a clinical trial [14]. Cameron and colleagues found that patient education on study procedures and patient rights were critical topics for clinical trial participants. Higher patient education level attained, prior experience participating in a clinical trial, and participant perceived understanding of clinical trials were factors significantly associated with clinical trial knowledge among clinical trial participants. To this end, the focus of the instrument development in this report is knowledge pertaining to clinical trials. Adequate patient health literacy is not only being able to utilize the necessary skill sets to navigate the healthcare system but includes applying knowledge comprehension to understand medical jargon [15]. However, healthcare providers and researchers commonly use complex terms, language, and medical jargon in their interactions with patients in delivering information about clinical treatments and other risks [16]. In addition, people with low education levels have been associated with low reading comprehension and health literacy levels [17]. A degree of working knowledge about clinical trials may enhance health literacy, serving as a foundational building block to build comprehension, self-efficacy, and skills to navigate clinical trials in a confident manner [18]. For these reasons, we hypothesize that higher education levels will be associated with a higher Clinical trial HEalth Knowledge and belief Scale (CHEKS) score. This study provides a summary of the development and evaluation of a survey to assess clinical trial knowledge and beliefs of clinical trials. Our purpose was to create a tool that assessed patient clinical trial knowledge and beliefs of observational and community-based research, as well as knowledge about safeguards in place to protect patient safety and rights, to identify how much people knew about the structures in place to overcome typical barriers to research engagement.

## 2. Materials and Methods

### 2.1. Survey Development Phase

Survey item development began with a review of the literature to identify salient issues and concerns around clinical trial research. The authors brainstormed key search terms in consultation with NYU faculty knowledgeable about clinical trial research, and survey design and evaluation. The lead author (A.C.) reviewed the United States Department of Health and Human Services, what are Clinical Trials and Studies? And the Mayo Clinic Clinical Trials websites to generate initial terms. PubMed and GoogleScholar were searched in the month of October 2017 for peer-reviewed literature to generate search terms. The authors identified key concerns that may influence a person’s decision to engage in a clinical trial, and developed survey items around these areas, such as safety, privacy, and the benefits of participant engagement. Survey item development was finalized by the survey developer, author (R.D.H.), and the lead author applied a community-engaged approach (See Figure 1 below) to iterate and vet survey items, before determining final items for inclusion. Once a consensus was reached, subject matter items for clinical trial knowledge items were determined: For example, because not all clinical trial studies are pharmacological, observational studies that are lifestyle interventions to improve health outcomes can take place in a community setting. Other subject matter items targeted patient safety, patient rights, the role of Institutional Review Board, and the role of people involved in the study to assess knowledge and beliefs identified as common barriers to clinical trial participation.

Search terms included: ‘clinical trials’, ‘clinical trial participation’, ‘barriers to clinical trial enrollment’, ‘clinical trial knowledge’, ‘clinical trial beliefs’, ‘clinical trial survey(s)’, and ‘clinical trial literacy’. This review found that there was a gap in the literature on validated survey instruments that assess clinical trial knowledge and beliefs. Additionally, we found that low clinical trial participation was a global issue. Research articles from India, Korea, Canada, and the United States reported low clinical trial participation rates stemming from mistrust of research, misperceptions about research, and lack of knowledge about the research process [12,13,14]. All newly developed survey items and responses underwent an iterative process of reviews and editing until a final list of 31 items were constructed (See Table A1 for the full measure). Seven expert research faculty at NYU Grossman School of Medicine, Department of Population Health, on clinical trial research provided feedback on survey item development and content validity. Expert faculty were asked to review the items for any content gaps and if the items represented knowledge and beliefs. Four iterations of survey item refinement were undertaken with the research team, Community Steering Committee, and expert research faculty, to generate and develop the questions among the experts. We pretested the survey items with our Community Steering Committee and research associates on the study team. The study design included a combined patient-centered approach guided by a community-steering committee for user appropriateness, combined with the commonly used MTurk crowdsourcing panel study design [19]. The study setting for the patient-centered approach was at NYU Langone Health and on MTurk for study recruitment. The objective of this study was to develop a clinical trial knowledge and belief measure and evaluate its validity.

### 2.2. Patient Centered Approach 

Health educators and community steering committee (CSC) members with prior clinical trial participation experience, provided feedback on survey items during their development. Health educators and community members were active members of their community board, activists, and public health advocates who were an integral part of the survey tool development to ensure that the items were relevant, and literacy level appropriate. The CSC reviewed survey items developed by the research team via focus group, iterated on survey item inclusion, and provided guidance on preferences to ensure privacy, confidentiality, and other safeguards to secure trust that their health information would be safe and protected. Their inclusion was essential to our patient-centered approach. Health educators and CSC members were recruited from referrals for the leader health educator at the Center for Healthful Behavior Change within the Department of Population Health, at NYU Langone Health. Iterative feedback on response options and level appropriateness was obtained to ensure survey items were tailored to patient needs. CSC members also provided input on survey length and content. 

### 2.3. Participants

Participants completed the survey items on the Amazon Mechanical Turk in November 2017 to gather responses needed to establish content validity of the survey measure. Amazon Mechanical Turk (MTurk), an online survey panel supported by Amazon, was used for participant recruitment (*n* = 409). The MTurk survey panel setting has been recognized for its low-cost and quick access to survey large samples of respondents in a rapid timeframe [20]. Participants were required to be able to read and understand English and be at least 18 years of age. Eligible participants proceeded to the online consent page via an online advertisement. After informed consent was obtained participants continued to complete the survey.

Demographic information was collected on a subset of the sample. The demographic questions asked were the following: “What is your gender? What is your race? What is the highest degree or level of school you have completed? Employed? What is your age?”.

The survey was estimated to take about 20 minutes to complete based on “think aloud” cognitive interviews with five adults [21]. Mturk participants were compensated $15/h, appropriated for the time needed to complete the survey. IRB approval was obtained in 2017 by NYU Langone School of Medicine (approval number: S17-00170). See Table A2 for a more detailed description of the sample. 

### 2.4. Analysis

We hypothesized that a subset of the items could be combined to create an overall scale. Hence, we examined item-scale correlations (corrected for overlap) for the 31 items, eliminated items with item-scale correlations less than 0.30, and then estimated internal consistency reliability for the remaining items [22] (See Table A3). In addition, we fit a categorical confirmatory factor analytic model to evaluate whether the items were sufficiently unidimensional. Because chi-square is too sensitive to sample size, we rely on the comparative fit index (CFI) and the root mean squared error of approximation (RMSEA) to evaluate model fit. 

To provide some initial information about construct validity, we estimated product-moment correlations of the simple-summated knowledge scale with age, gender, and education. We estimated a one-way ANOVA to assess the association of race with the knowledge scale score. We hypothesized that the CHEKS would be positively associated with age and education scores. We also hypothesized that African-Americans would have lower CHEKS scores than Whites. 

Finally, we fit an item response theory graded response model and obtained item characteristic curves to evaluate whether the CHEKS response options were monotonically associated with the underlying concept of knowledge and beliefs (See Figure A1). 

## 3. Results

### 3.1. Sample and Evaluation of Survey Items 

The study sample size (*n* = 409) and was predominantly young adult White and Asian, low-middle income, and employed. The average age of the sample was 34 (SD = 15.7); 48% were female and 35% were White, 11% were Black or African American, 44% were Asian, and <1% were Native Hawaiian or Pacific Islander. Among participants, 56% had a bachelor’s degree, and 79% were employed. Twenty-six percent reported an income between $20,000 to $39,000 and 20% reported an income between $40,000 to $59,000. 

We initially began our survey with 31 items, and identified 6 of the 31 items that had item-scale correlations (corrected for overlap) lower than 0.30, that were omitted from the survey. The final survey items are in Table 1 below, and the full list of all 31 survey items that we began with can be found in the Appendix A. The survey response options were (4 = Definitely True, 3 = Somewhat True, 2 = Uncertain, 1 = Somewhat False, 0 = Definitely False.). Coefficient alpha for the remaining 25 items was 0.93. A one-factor categorical confirmatory factor analytic model, including factor loadings and standard errors, with 16 correlated errors was not statistically significant (chi-square = 791.610., df = 239, *p* < 0.0001) but fit the data well (RMSEA of 0.07 and CFI of 0.95). Internal consistency reliability of the 25-item CHEKS was 0.93 and ordinal alpha was 0.95. We recoded each item linearly to have a 0–100 possible range and averaged together the 25 items to produce the CHEKS scale score. The mean score was 75 (range 15–100, SD = 15).

Item characteristic curves from the graded response model are shown in Figure A1. These curves show estimated scale z-scores on the x-axis plotted against probability of responding on the y-axis. The response categories functioned well overall (i.e., monotonic association between response options and probability of response across the continuum), but the second response option was never most likely to be selected for items 10, 13, 20, 24, and 31.

### 3.2. Associations of Clinical Trial Knowledge with Demographic Variables

The CHEKS was not significantly associated with patient gender. As hypothesized, the CHEKS was significantly positively correlated with education (r = 0.15, *p* = 0.0365) and patient age (r = 0.20, *p* = 0.0051). The overall F-statistic for the association of race with CHEKS was significant (F = 2.73, *p* = 0.0207), but the Duncan multiple range test indicated no significant pairwise race subgroup differences.

## 4. Discussion

In the current study, we developed and evaluated the CHEKS tool for assessing clinical trial knowledge and beliefs. We confirmed our hypothesis that level of education, age and race were significantly associated with CHEKS scores. This is important, given that health knowledge has been established as a predictor of individuals’ ability to process and evaluate health information [23]. Assessing patient health knowledge in the context of clinical trials ensures that each participant has a level of knowledge and skills needed to understand and use information that will empower them to make an informed decision regarding participation [24]. Understanding patient beliefs may help to address and overcome misconceptions around clinical trial research, that may impact participant participation.

Our patient-centered study design was a critical factor in developing question stems and response categories. The participant preferences related to troubleshooting, data privacy and confidentiality, data storage location and collection, and user access were an integral component of participant engagement with clinical trials on digital devices [25]. Questions for the CHEKS tool were guided by our CSC, expert research faculty and the literature. Participants’ rights and safety, as well as the process and procedures of clinical trial research, funding, setting and benefits, were identified as critical areas to ensure patient understanding. These topic areas were agreed upon as survey items to include in the CHEKS tool given the historical mistrust of research and typical misconception of clinical trials only being pharmaceutical in nature [11,12,26]. Our approach is consistent with the literature, with our patient-centered input to the instrument development for key content areas guided by the literature, literacy level and cultural appropriateness guided by the CSC, data collection, troubleshooting user access processes, and survey ease of use administered online. 

Finally, financial incentives offered through mTurk, was an effective motivator for a high response rate, despite the 31-item survey tool length [27,28]. Questionnaire length did not affect response rate, but financial incentives were positive motivators. This finding is consistent with our monetary compensation to mTurk panel users of $15 USD/h. However, a more efficient shortened instrument without taking away its reliability may be a way to reduce participant burden. Mturk has been known to be useful for collecting a large amount of data quickly on more racially diverse populations, including more non-White participants [29]. However, in this study, the majority (44%) of respondents identified to be of Asian descent and there was no representation of Hispanic or Latinos. There was also a low percentage of Blacks, making our study less diverse than intended. Future research evaluating CHEKS among Hispanic and Black populations, may aid in overcoming barriers to research engagement, among a group who are largely not represented in research, yet disproportionately affected by health disparities.

This is an initial study to create and evaluate the CHEKS. Further research is needed to provide more information about reliability and validity. Further refinement of CHEKS to inform its reliability and validity could help ascertain patient reluctance to research, and aid in supporting clinical trial participation. Increased clinical trial participation, may help in developing health solutions for diverse populations. CHEKS has the potential to ascertain patient knowledge and belief gaps about clinical trial research, and aid researchers to develop targeted messaging and communication strategies to overcome these gaps, to optimize clinical trial literacy for patients to make informed decisions about engagement. This tool could help with increasing clinical trial engagement, enrollment, and retention via enhancing clinical trial literacy. 

### 4.1. Implications for Future Research and Practice

Study implications for communication, practice, and research include leveraging this measure as a needs assessment or baseline intake tool of participant knowledge and beliefs about clinical trials to identify knowledge gaps and beliefs held about research. To this end, community health workers, public health researchers, health behavior scientists, population health researchers, and observational study and community-engaged researchers could potentially benefit from this tool. The CHEKS tool would be largely disseminated via national public health societies, and promoted on social media channels with partner organizations who share similar goals of increasing clinical trial knowledge, beliefs and literacy to optimize patient engagement, and encourage its use. This tool can be administered as a survey to gauge people’s knowledge, beliefs and level of awareness, to create interventions and/or educational materials to counteract misperceptions about clinical trials research. The target audience for CHEKS is patients and research study participants in healthcare research, prevention, or treatment studies. Since there are age restrictions on some clinical research, this tool is geared towards people 18 years and older. In addition, our measure could be used as part of pre/posttest research analysis to determine how the research community is faring in improving recruitment and participation in trials from all communities. Examples of such interventions include the development of a tailored website to increase participant willingness to enroll in clinical trials [11], to understand knowledge content areas to bolster and beliefs or misconceptions to clarify. 

Next, future studies in the U.S. are needed to evaluate the generalizability of the CHEKS measure. Our measure offers one item related to the inclusion of people of color. While limited, future studies could expand on this item to include a measure of clinical trial knowledge and beliefs, tailored and tested among BIPOC populations. This is particularly important because the tool used in this study was only available to respondents in English. Typically, when English is not your first language, it makes reading and comprehending medical jargon more daunting [30]. Future research may consider translating and evaluating CHEKS in different languages, which would present an opportunity to enhance communication between researchers and potential participants. Additionally, doing so could foster more collaboration between global health researchers and may improve research quality. 

### 4.2. Limitations and Future Research

There are several limitations to using the MTurk online survey panels as a recruitment schema. First, data integrity is an issue, such that respondents may develop mechanisms to complete surveys quickly to receive the proposed incentive [31]. As such, participants are limited to those who can access online surveys through the MTurk platform, and may provide undesirable responses including, providing the same responses to the same questions, answering questions too quickly, or providing false answers [32]. As indicated in our limitations section, participants may have been in acquiescence due to the overwhelmingly positive responses on survey items, which may threaten data integrity. Given the low representation of Black Indigenous People of Color (BIPOC), among survey participants, the responses may not be representative of BIPOC communities, who have the greatest need for tailored measures on clinical trial knowledge and beliefs given their low representation in clinical trial research. Also, our survey was developed based on research in the United States, and survey items may not be applicable in other context or countries. Additionally, all of the survey item responses were in a positive response direction, with no negative response options. Including negative response options in future iterations of the survey, the tool may help to better ascertain knowledge versus participant acquiescence. 

The Hawthorne effect is a well-documented phenomenon that affects many research experiments in the social sciences [31]. It describes the bias the participant may having knowing they are being observed in a study. By using MTurk, our participants had the intention to take our survey and be compensated. This may have changed some of their behaviors when answering questions. Future studies should include diverse sample size and culturally responsive question items that are reflective of the concerns of a racial and ethnically diverse population.

## 5. Conclusions

Assessing clinical trial knowledge and beliefs can provide useful information about patient understanding, and lead to targeted communication initiatives focused on addressing knowledge gaps and belief misconceptions. Overcoming these knowledge gaps and misconceptions could contribute to larger engagement in clinical trial research, and ultimately, more diverse racial and ethnic participants. This tool has the potential to benefit public health researchers, behavioral scientists, and populations health researchers, particularly given the focus of this tool to focus on observational and community-informed research, going beyond typical oncology and pharmacological clinical trial measures. Clinical trial knowledge, including understanding the process, procedures, and patient rights related to clinical trials is the first step to addressing participant concerns that may serve as barriers to enrollment. The evaluation of the Clinical trial HEalth Knowledge and beliefs Scale (CHEKS) in this study is the first step to assessing patient clinical trial knowledge and honing in on key knowledge gaps to address. CHEKS serves as a promising tool to assess clinical trial comprehension that could optimize patient participation in research.

## Figures and Tables

**Figure 1 ijerph-19-08660-f001:**
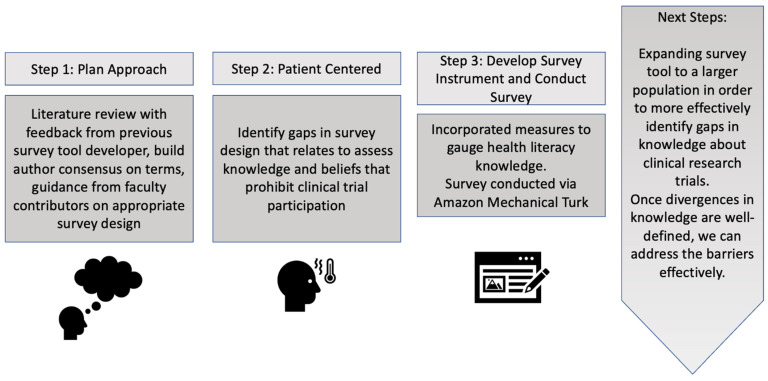
Clinical Trial Health Knowledge and beliefs Scale Development Process.

**Table 1 ijerph-19-08660-t001:** Final survey items included in the Clinical Trial Health Knowledge and Beliefs Scale (CHEKS).

Variable Name	Item Stem
Q1	A clinical trial is a research study that involves people.
Q2	Clinical trials test strategies designed to improve health.
Q3	An intervention is a new treatment or strategy that is being tested by the research team.
Q4	The goal of a Clinical Trial is to find out if an intervention works.
Q5	The goal of a clinical trial is to find out if an intervention is safe.
Q7	A research team is led by a principal investigator.
Q8	Clinical trials can be funded by universities.
Q9	Clinical can be funded by private companies.
Q10	Clinical Trials can be funded by the government.
Q11	Clinical Trials can take place at a hospital or doctor’s office.
Q12	Clinical Trials can take place in a doctor’s office.
Q13	Clinical Trials can take place in my community.
Q14	Clinical trials follow a research plan called a protocol.
Q15	The research plan is explained to the participants before the start of the study.
Q16	The risks of research are explained to the volunteers before they agree to take part in the research study.
Q17	The potential benefits are explained to the volunteers before they agree to take part in the research study.
Q18	Participation in a clinical trial is voluntary.
Q19	You can choose to leave the research study at any time.
Q20	Research participants are kept updated as the study goes on.
Q23	Patient medical information is kept private during a clinical trial.
Q24	The institutional review board exists to protect the patient’s rights during a clinical trial.
Q25	The institutional review board is an ethics group that reviews the research plan before the start of a clinical trial.
Q26	If a clinical trial is found to be unsafe, the research study will stop.
Q27	If an intervention is found to be unsafe during a clinical trial the intervention will be discontinued.
A31	Research is important to improve the health of people of color.

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
