# Peer review of "Development and Evaluation of the Clinical Trial HEalth Knowledge and Beliefs Scale (CHEKS)"

_ijerph, 2022, doi:10.3390/ijerph19148660_

Round 1
Reviewer 1 Report
Thank you for the opportunity to review your manuscript. I agree that health literacy is an important area to focus on and developing a tool to understand people's knowledge and beliefs about clinical trials is useful. However I have a few comments which I hope are helpful.
Overall comments:
I felt that you didn't clearly articulate how health literacy impacts clinical trials. For example does it impact recruitment to trials, does it impact retention? I think the introduction should clearly articulate why you focused on clinical trial knowledge and beliefs and why this is important to do.
The results are very statistics heavy. I am not qualified to comment on the appropriateness of statistical methods as this is not my expertise but as a qualitative researcher who is genuinely interested in the paper, I could not follow what the statistics indicated and what the tool did. There was a focus on removing overlapping outcomes but I didn't really get a sense of what your tool showed and how the results of the tool could be used. For example, is it to help show what the public doesn't know about trials so that this information is presented differently or focused on?
Throughout there is some inconsistency in terminology - measure, tool etc - it would be helpful to explain what you mean by the terms and use it consistently.
There are also quite a few typos in the paper. E.g. generalizable should be generalizability. Missing full stops in places.
What does CHEKS stand for? I couldn't see how you got this acronym from the words it relates to.
Specific comments:
Abstract - add a sentence about why patient health literacy is vital to clinical trial engagement. I know abstracts with limited word count make it difficult but I think the background/justification for your study is missing (in my opinion). Could you add number of participants and year of study in abstract?
Introduction
I think you should define health literacy in the introduction. Also health literacy is not only about knowledge but other skills and this should be brought in to the introduction. Health literacy can also be addressed by health organisations by making information more user-friendly.
Para 1 - last 2 sentences could be moved to para 2 and the 1st 2 sentences of para 2 could be moved to para 1 to improve flow.
Can you explain why health literacy is important and how it impacts trials in the introduction?
You say no validated tools for health literacy about clinical trials exist but I think it would be helpful to show what does exist and perhaps a quick summary of health literacy tools for other research areas. you mention cancer trials but are there other tools available and were these useful for when you were developing your tool? some of the information in section 1.1 could be moved to the introduction main section again to help flow of information.
You say that the tool is for clinical trial knowledge but later you say also beliefs (and in the title) - just needs to be consistent.
I was confused that you said that the tool was to look at knowledge of CTs but then you mentioned observational and community-based research - can you explain what you mean here?
Methods
2.1 Search terms - I think you focused on US sites but I felt that methods needed more detail. Did you search databases for peer-reviewed literature and if so which databases did you use? Some of the sentences either need to move into the introduction or reframed to reflect methodological decisions made.
I was unclear how the survey items came from the literature. can you explain the process of finding potential items, how they were reviewed, whether any re-wording occurred. At the moment, I don't think someone could replicate your study.
Fig 1 seems to show a different process to the text (e.g. patient centred activity comes first in the figure but seems to have been later according to the text.
2.3 Measures - are you using 'measures' to refer to survey items? I wasn't sure
2.1 You mention knowledge and beliefs but introduction only says knowledge.
2.2. patient-centred
I felt more detail could have been included here about activities and impact. some new information about this in included in the discussion without it specifically being mentioned in the other parts of the paper.
2.4 Participants. Can you start with who you were aiming to recruit and your strategy? I am not familiar with Mturk - it may be useful for readers for you to explain a bit more about this. Suggestion to add months and year that study was undertaken as Fall is a US term.
Suggestion: add a section in methods about collection of demographic data (separate section in methods)
2.5 Analysis
Were decisions made a priori?
Tense moves from past to present.
Do you need the hypotheses here? some are presented elsewhere - not all presented together
Results
Participants - methods says n=409 but results say n=207. Is 409 who you approached and 207 took part?
What are the survey items? You state you had 31 items but I couldn't see these listed anywhere. You refer to the 25 you kept in the appendix but I think there should a be a list of all of them.
How did people use the tool? Is it yes/no, or agree/disagree/not sure etc.
I didn't get a sense of how people completed it. Is the idea that a total score is calculated for each participant? Are there cut offs for what is considered 'good' knowledge/beliefs about clinical trials?
I couldn't see what the tool achieved - it showed different responses based on a participant's race. age and education - but did you use the tool to see if it could do this? It was unclear (for me)
Discussion:
Tense should be consistent.
Is this a valid tool? Is this study more of developing and piloting a tool rather than validating it? This feels like an initial study that needs more work to validate it - which you refer to in the discussion but I think you need to explain the impact or potential impact of this tool in more detail. How will it be used? how will it help clinical trials and researchers? How will people know about your tool and how will you encourage its use? You touch on some of this but not in any specific detail and I think that would strengthen the paper.
Limitations - you only obtained results from people who can access online surveys - this excludes people who do not use computers/have internet etc - this population may have completely different health knowledge and I think you need to include this as a limitation. It has also only been tested in one country so you should include a statement about how relevant this tool would be in other contexts/countries
Reviewer 2 Report
The study involves development and validation of Clinical Trials Health Knowledge and Belief Scale. The manuscript needs some minor edits. The method and results need to be better illustrated.
Here are my comments:
-
Please revise the introduction as the purpose of the study was stated multiple times. It is best to consolidate the purpose at the end after making a strong case.
-
Please add all reference lines 118-119.
-
Line 134-135: this needs to be in the introduction not in the method.
-
Figure 1, block 2 : Did you mean developer ? not development?
-
Page 4, paragraphs 2.2 and 2.3 have the same exact content
- In general the method needs to be better explained. The validity assessment are not all clear.
-
Line 166: how does taking the survey establish face content validity? Please include the details on this process.
-
Face and content validity are 2 separate entities. Please describe how each one was assessed.
-
Please show all data mentioned in the analysis section in tables : Cronbach’s alpha, CFA, RMSEA
-
Were the values standardized ?
-
Based on the 10:1 ratio used as a rule of thumb, the sample size is small for IRT analysis.
-
It was not clear how reliability was assessed, please provide more details.
-
The discussion is very short. You can discuss the choice of questions, how these questions will help take care of the health literacy issue, what would be the impact on future clinical trials, etc.. In essence, the discussion needs to convince the reader that this new tool is well thought-out and is a useful one.
-
Please add some explanation to Figure A1 explaining how to interpret these results for readers not familiar with Item Characteristic Curves.
Round 2
Reviewer 1 Report
Thank you for making revisions to your paper. It is much easier to read and absorb now and I appreciate the efforts put into these changes. I have a number of small comments/suggestions that I think are needed:
- can you please add date of searches in section 2.1
- table A2 - suggest changing to 'participant characteristics'
- I appreciate your comments about not wanting to confuse people by including the six survey items that were ultimately removed but I think this information should be available to readers. My suggestion is you include within your main manuscript a table with the final survey items and in the supplementary files, the table with all survey items that you began with. This way it's clear that people know what the final tool is but if interested can see what you began with.
- I appreciate you adding clarification of the survey response options in the supplementary file (4 = Definitely True, 3 = Somewhat True, 2 = Uncertain, 1 = Somewhat False, 0 = Definitely False.”) but I feel that this should be in the main manuscript in the methods section
I cannot comment on the stats side of things but understand you have reviewed the feedback on these from another reviewer.
Reviewer 2 Report
Thank you for the addressed concerns.
After reviewing the authors' responses and the edited manuscript, I don't think using the term "validity' in the title is appropriate. It implies that the scale has been validated and ready for use. There are many other validity tests that were not performed.
I was hoping that the authors provide more information to support its use in its current form. There is a lot to be done to validate this questionnaire. The fact that the authors chose not share more information about the CFA ( loading factors, scree plot, variance, etc..) , it is hard for the reviewer to confirm construct validity ( which is the most important validity).
My suggestion is to change the title to : Development and evaluation of ....
Also for table 3, please specify the p value used.
